# Mediterranean Dietary Pattern and Cardiovascular Risk in Pregnant Women

**DOI:** 10.3390/life13010241

**Published:** 2023-01-14

**Authors:** María Morales Suárez-Varela, Isabel Peraita-Costa, Alfredo Perales Marín, Beatriz Marcos Puig, Agustín Llopis-Morales, Jose M. Soriano

**Affiliations:** 1Area of Preventive Medicine and Public Health, Department of Preventive Medicine and Public Health, Food Sciences, Toxicology and Legal Medicine, School of Pharmacy, University de Valencia, Avda. Vicent Andres Estelles s/n, 46100 Valencia, Spain; 2Biomedical Research Center Network on Epidemiology and Public Health (CIBERESP), Institute of Health Carlos III, Avda. Monforte de Lemos, 3-5, Pabellón 11, Planta 0, 28029 Madrid, Spain; 3Department of Gynecology and Obstetrics, La Fé University and Polytechnic Hospital, Avda. Fernando Abril Martorell, 106, 46026 Valencia, Spain; 4Food & Health Lab, Institute of Materials Science, University of Valencia, 46980 Valencia, Spain; 5Joint Research Unit on Endocrinology, Nutrition and Clinical Dietetics, Health Research Institute La Fe-University of Valencia, 46026 Valencia, Spain

**Keywords:** nutrition, pregnancy complications, primary prevention, cardiovascular disease, risk factors

## Abstract

Nutrition during pregnancy is one of the most important factors that determine the health of a mother and the proper development of her fetus. The main objective of this study was to analyze the association between adherence to a Mediterranean dietary (MedDiet) pattern and cardiovascular (CV) risk factors in pregnant women. Accordingly, we carried out an observational, population-based study using data from pregnant women present in a hospital during the entire course of their pregnancy. Adherence to the MedDiet was assessed using the MedDiet score questionnaire. Our study identified that 87.25% (95%CI: 83.48–90.27) of the women had a cardiovascular risk in relation to their dietary intake. Women with diet-related CV risk were more likely to smoke (*p* = 0.004), weighed more at the beginning of pregnancy, engaged in little physical activity, and had lower adherence to the MedDiet pattern than women without a diet-related CV risk. Dietary analysis showed low consumption of cereals, vegetables, and fish, which failed to satisfy the recommended portions in Spain. Adequate adherence to the MedDiet was found for 54.2% of women who were considered to be without CV risk and 45.8% of women with CV risk. Our data suggest that the MedDiet could be improved in relation to the consumption of cereals, vegetables, and fish during pregnancy in order to reduce CV risk.

## 1. Introduction

Pregnancy is a physiological stage effecting changes in nutritional requirements, thereby increasing energy and nutritional needs [1]. Maternal nutrition before conception and during pregnancy is considered one of the most influential factors affecting the proper growth of the fetus and the possible future development of chronic diseases during in adult life [2]. Poor nutritional intake is associated with birth defects, intrauterine growth retardation, premature birth, and miscarriage [3]. In contrast, high energy intake is associated with increased maternal weight gain, macrosomia, and increased risk of cesarean delivery [4]. For all the aforementioned factors, food and, more specifically, an adequate dietary pattern together constitute a fundamental pillar in the prevention and prognosis of CV diseases [2,4].

There are different epidemiological studies considering nutrition through a more holistic approach that focuses on the role of dietary patterns rather than individual nutrients, foods, or food groups [5,6,7,8,9,10], as this approach is more likely to better reflect long-term eating habits and behaviors. Among dietary patterns, the Mediterranean Diet (MedDiet)—a diet rich in vegetables, fruit, whole grains, legumes, nuts, and olive oil; incorporating a moderate intake of dairy products, fish, chicken, and wine; and including low consumption of red meat [11]—seems to be one of the most widely accepted diets as due to its positive effects on human health [12,13,14].

The literature supports the numerous protective and preventive effects that the MedDiet pattern exerts on chronic non-communicable diseases [15]; despite this, we are facing a nutritional transition away from this pattern in the Mediterranean countries due to the substitution of carbohydrates for other sources of macronutrients [16]. Currently, the average diet in Spain shows a progressive departure from the MedDiet pattern [17,18].

The MedDiet pattern is associated with benefits during pregnancy and the prevention of certain diseases, including CV diseases [19,20,21,22,23]. Components of the MedDiet pattern have been shown to be effective in reducing CV risk and have been associated with lowering blood pressure, glucose levels, triglycerides, and total cholesterol when these nutrients are included in a dietary pattern [2,4]. It has been proven that adherence to this pattern produces a decrease in plasma glucose, blood pressure, oxidation of LDL cholesterol, and inflammatory markers, and an improvement in the lipid profile [24]. The potential effect of the MedDiet with respect to improving endothelial dysfunction and reducing CV risk could be attributed to its fatty acid profile, as lipids—mainly cholesterol and saturated fatty acids—are the compounds most involved in endothelial dysfunction [25,26].

According to the World Health Organization (WHO) [27], in Spain, 28% of all deaths occur because of chronic diseases, specifically, CV diseases, and 5% of deaths occur due to maternal, perinatal, and nutritional conditions.

Regarding the development of these maternal complications, there are several factors that are not modifiable, such as age, sex, ethnicity, or family history. Meanwhile, some studies have shown that nutritional factors might influence the risk of developing hypertension and other CV diseases during pregnancy [28]. Among the nutrition-related factors, obesity is a modifiable cause that can be changed through diet and, together with the combination of physical exercise, it can cause an improvement in maternal and fetal conditions and contribute to optimal weight gain [29]. Obesity is a problem that continues to grow; for instance, in 2015, it was estimated that the prevalence of obesity among pregnant women was approximately 30%, mainly in Western countries [30]. Maternal obesity is associated with numerous complications, such as hyperlipidemia, hyperleptinemia, hypertension, insulin resistance, micro or macroproteinuria, and endothelial and vascular dysfunction [31].

Inadequate weight gain or pre-pregnancy obesity causes adverse consequences during a child’s infancy, since it correlates with a high probability of increased blood pressure, inflammatory markers, imbalances in the lipid profile, and increased risk of obesity and insulin resistance [30]. In addition, the presence of gestational diabetes produces epigenetic changes in the intra-uterine environment that may increase susceptibility to the development of chronic noncommunicable diseases that affect the adult stage of the child [32]. The development of gestational diabetes is related to an increased risk of type II diabetes mellitus after pregnancy [33].

Although previous studies have analyzed the positive effects of the MedDiet pattern on the health of the population, we have not found a relationship between the MedDiet and CV diseases during pregnancy. The aim of this work is to determine the association between adherence to the MedDiet and CV risk factors in pregnant women. We hypothesized that greater adherence to the MedDiet, defined using an a priori dietary pattern approach and by calculating the validated MedDiet score (MedDietScore) [34], would favorably influence the curtailment of CV risk.

## 2. Materials and Methods

All procedures were in accordance with the ethical standards of the Clinical Research Ethical Committee of the La Fé University and Polytechnic Hospital (Ethic Committee N° 2014/0116 of 8 September 2014), which included a confidentiality agreement concerning the data collected according to Organic Law 15/1999 of 13 December regarding the Protection of Data of an Official Nature and in accordance with the 1964 Helsinki declaration and its later amendments or comparable ethical standards.

This study is an observational, population-based, periodic, cross-sectional study using data from women who had been attended to throughout their pregnancy (from pregnancy’s confirmation before 12 weeks and with at least one visit per trimester) at the Valencia-La Fé Health Department and gave birth at the La Fé University and Polytechnic Hospital of Valencia (Spain), during 2020 and 2021.

The Valencia-La Fé Health Department includes: the La Fé University and Polytechnic Hospital, the Ernest Lluch Campanar Health Center, the Ricardo Trénor Palavicino Specialty Center, and 20 primary care centers (12 health centers and 8 auxiliary offices).

La Fé University and Polytechnic Hospital is a public hospital, and part of the National Health Service, that serves as the reference hospital of the Comunitat Valenciana. The hospital is responsible for the health care of 300,000 inhabitants and added to this population are patients from other health departments who come to the hospital due to its reference status. The hospital attended to over 4000 (4307 and 4126) births in each of the years this study was carried out.

Our sample size calculation with a confidence level of 95%, a margin of error of 5%, a population proportion of 50%, and a population size of 8000 resulted in a required sample size of 367 women.

For this study, in addition to having been attended to throughout pregnancy at the Valencia-La Fé Health Department and given birth at the La Fé University and Polytechnic Hospital, the participants were required to have met the following established inclusion criteria: given live singleton birth, had electronic medical records from at least 4 different consults during the antenatal (one per trimester) and perinatal (at or within 5 days post birth) periods available, and had data regarding personal characteristics, habits, and dietary intake available. Women with multiple pregnancy or fetal loss at any time during pregnancy up to and including birth were excluded. In the event that a woman gave birth twice during the study period and was eligible for inclusion on both occasions, only one pregnancy would be considered.

To ensure that the sample size threshold was met, a group of six hundred eligible women were identified and offered participation during a post-birth visit. Four hundred (66.66%) women gave informed consent to be included in the study. Written informed consent was obtained from all individual participants included in the study. The recruitment process is shown in Figure 1.

Data on personal characteristics and habits as well as dietary intake were retrieved from information collected through a face-to-face interview during the participants’ last programed antenatal visit (36–38 weeks) to reflect their habitual diets during the entirety of their pregnancy. A review of all clinical history was also performed to retrieve all relevant anthropometric and medical data.

### 2.1. Personal Characteristics

Different information was collected from the participants, such as age (years); country of origin; weeks of gestation; parity; family history; presence of diabetes; consumption of or exposure to drugs, alcohol, and tobacco; and intake of vitamin and vitamin supplements. 

Regarding the anthropometric data, height, weight, and BMI prior to pregnancy; weight; BMI in the third trimester; and total weight gain were recorded. Preconception weight was taken from the latest available medical records and total pregnancy weight was obtained from clinical records composed during the follow-up period. According to the Institute of Medicine Guidelines, a weight gain of 0.44 to 0.58 kg/week for women with a pregnancy BMI of less than 18.5; 0.35–0.50 for women with a pregnancy BMI of 18.5–24.9; and 0.23–0.33 kg for women with a pregnancy BMI of 25.0–29.9 is suggested. That is, the recommended weight gain values at the first trimester for women with a pre-pregnancy BMI of <30 and women with a pre-pregnancy BMI of >30 are 2 and 1.5 kg, respectively [35].

Information regarding maternal education (no formal education; completion of primary studies, secondary studies, and tertiary studies), place of residence (rural or urban), and employment status (employed or unemployed) during the months of pregnancy were retrieved.

Level of physical activity was self-reported and collected using a questionnaire that reproduces the corresponding questions from the Spanish National Health Survey, which, in turn, are based on the International Physical Activity Questionnaire (IPAQ) [36], which establishes three levels of physical activity: Category 1: Low, Category 2: Moderate, and Category 3: High. An additional level, Category 0: None, was also included for completely sedentary individuals.

### 2.2. Dietary Assessment

To assess dietary intake, the Mediterranean diet score (MedDietScore) [34] questionnaire was administered. It is used to assess adherence to the MedDiet and estimate only diet-related CV risk [34].

Women were instructed to complete the MedDietScore questionnaire, validated for the general adult population, once, which occurred during their last programed antenatal visit (36–38 weeks), in order to reflect their habitual diets during the entirety of their pregnancy.

Adherence to the MedDiet was measured using a scale (0–55), with higher values indicating greater adherence to the MedDiet. The consumption of 11 food groups was determined: unrefined cereals, vegetables, fruits, legumes, potatoes, chicken, red meat and derivatives, fish, dairy products, the use of olive oil, and alcohol. The values 0, 1, 2, 3, 4, and 5 were assigned for the consumption of foods that are positively associated with this pattern; these values correspond to never, rarely, frequent, very frequent, weekly, and daily. On the other hand, for dairy products, red meat, chicken, and alcohol, the scoring scale was reversed, since it is negatively related to this dietary pattern [34].

MedDietScore distributes the different values into five categories (0–11; 12–22; 23–34; 35–44; 45–55) of adherence and, based on the score achieved, estimates CV risk. An individual with optimal adherence (44–55) serves as the reference; individuals with scores 35–44 have a CV risk of 1 (equal to the optimal adherence group); scores 23–44 present a risk of 1.42; 12–22 a risk of 1.65; and 0–11 a risk of 2.17.

### 2.3. Statistics

Data are presented as mean ± standard deviation for continuous variables and were compared via Student’s *t*-test. Categorical variables were evaluated with the Chi-square test and Yates correction, while differences between categorical and several clinical and nutritional variables were tested using non-parametric Mann–Whitney U test. Comparison between various variables and terciles of MedDietScore were performed using Kruskal–Wallis test, and Bonferroni correction was used to account for the increase in Type-I error. All statistical tests were two-tailed and a *p*-value < 0.05 was considered significant. The statistical analysis was performed with the SPSS statistical package, version 23.0.

## 3. Results

### 3.1. Sociodemographic Characteristics

As shown in Table 1, the pregnant women had a mean age of 32.5 ± 5.71 years, half of them were nulliparous, their weights before pregnancy were around 65 Kg, and the average height was 1.63 ± 0.06 m. Most of the women came from Spain (311) and the rest were of American, African, and Asian origin.

The duration of the pregnancy was approximately 39 weeks, the mean weight gained was 12 ± 8.44 Kg, and the BMI in the third trimester was 29, for which the latter was somewhat higher in women with a lower CV risk.

Approximately 80% of the participating women performed physical activity during pregnancy, although it was mostly of a light intensity. In contrast, moderate activity was more frequent in the group of women with no CV risk.

Folic acid intake was higher in women at higher risk, being common in 71.5% compared to 54% in the other group. The consumption of vitamin supplements was also higher in this group, although not much of a difference was seen between the two groups. Iron was taken supplementarily by 82% of the women. It can be seen that almost none of the women used food supplements.

Of all the variables analyzed and identified in Table 1, only tobacco consumption was significantly different between both groups.

### 3.2. Cardiovascular Risk

Of the 400 women interviewed, 87.25% had CV risk while the remaining 12% did not have such risk. In general, women with CV risk are more likely to be smokers, have a higher pre-pregnancy weight, suffer from gestational diabetes more frequently, and spend less time engaging in physical exercise than women without risk. The estimation of the probability of possessing very-low CV risk for the participants is between the ranges 0–11 and 12–22. This aspect is positive since these values represent the minimum adherence to the MedDiet.

Within the group of women with CV risk, 53.7% of them have a 1.42 probability of suffering from this risk and 45.8% have a 1 probability. In the group of women classified without CV risk, 43.9% have a 1.42 probability of being at risk and 54.2% have a 1 probability. A higher score in the 35–44 range can be observed in the participants without CV risk.

None of the participating women reached values of 45–55, that is, the maximum adherence to the MedDiet.

The different scores of the scale that was used in the MedDietScore program for the evaluation of the diet are discussed below.

Scores of 35 and over are considered to correspond to good adherence to the MedDiet. Of those women who exceed this value, 12% had a 70.9% level of adherence to the MedDiet; 13% had a 69.1% adherence level; 12% had a 67.3% level of adherence; 22% had an adherence level of 65.5%; and 30% of the participants had an adherence level of 63.6%.

### 3.3. Food Consumption Corresponding to Mediterranean Dietary Pattern

As shown in Table 2 and Figure 2, the consumption of unrefined cereals in the participating pregnant women stands out for covering a frequency of 5–8 times/month. In this range, the intake levels are higher, presenting values of 51.6% for pregnant women at risk and 43.5% for pregnant women without risk. Daily, only 1.2–4.3% of pregnant women eat unrefined cereals.

In general, the consumption of potatoes during pregnancy is low, since women ate this food between 1–4 times/month. At this frequency, it was consumed by 67.1% of the women at risk and by 57.4% of the women without risk.

A total of 32.9–38.3% of women had eaten fruit 9–12 times a month during pregnancy. A higher rate of consumption by women without CV risk can be clearly observed. Only 10% of the participants ate fruit daily, and between 15.6–23.4% did so weekly.

The consumption of vegetables stands out for being higher in women without cardiovascular risk, being eaten between 5–8 times throughout the month by almost 40% of these women and by 36% of women with CV risk. Between 4.3–6.9% of the participating population only ate this food group weekly and a very low percentage (0.5%) daily.

Both groups of pregnant women ate legumes with a frequency of 5–8 times during the month, and the intake of this food source was higher—at 67.7%—among women with CV risk compared to 59.6% of women without risk. Approximately 20% of all the interviewed participants consumed legumes between 9–12 times a month.

A total of 5.2–6.4% of women did not eat fish during pregnancy. The frequency of habitual consumption was between 5–8 times during the month, and this was incorporated in the diet by 61.4% of the women with CV risk and by 57.4% of the women who do not suffer from said risk. The weekly consumption of this food was very low for both groups of women, not exceeding 2%.

It can be seen that most of the pregnant women ate red meat and meat products between 1–4 times a month. The level of consumption does not exceed 5% for those women who added these products more frequently to their diet. Compared with the consumption of red meat and other meat products, it can also be seen that the largest share with respect to the consumption frequency of chicken is 1–4 times a month. However, this frequency differs in that approximately three times as many pregnant women incorporated chicken into their diet between 5–8 times throughout the month.

The intake of whole milk and other dairy products is higher in women with CV risk. It can be verified that more than 50% of the participants ate this food group daily and that 25% of the pregnant women ate it weekly.

It is necessary to highlight that olive oil is the main source of fat in the examined diets; it was consumed daily by 87.2–92.5% of the participants.

Table 3 shows a comparison of the frequency of food consumption according to the Spanish Society of Community Nutrition (SENC) recommendations [37] between the two groups of women, which were classified according to CV risk.

Regarding the intake of unrefined cereals, almost 10% of the women interviewed did not comply with the recommendations proposed by the SENC, and the values were similar among them.

It should be noted that 58.6% of the women with CV risk consumed between two to three pieces of fruit per day and that consumption was 72.3% higher in those participants who did not present this risk.

It is necessary to mention that the SENC recommends consuming two daily servings of protein-rich foods, alternating between fish, meat, eggs, poultry, and legumes [38]. By determining the consumption of this food group in this way, it has not been possible to individually identify whether this stipulation is met with respect to the recommendations for those foods. However, it can be shown that the least-consumed protein-rich food by pregnant women is fish, since between 5.2–6.4% of the participants never ate it during this period. Regarding the consumption of fish, the Spanish Agency for Consumption, Food Safety, and Nutrition (AECOSAN) advises that varieties with a low mercury content be chosen due to this element’s toxicity [39]. It is recommended that the intake of red and processed meat is moderate during this period since it has been verified that most of the participants opted for this type of meat. It is preferable to eat white and lean meats since they provide lower concentrations of sodium, cholesterol, and saturated fat [37] and, therefore, would be associated with a decreased risk of cardiovascular disease, obesity, diabetes, or other diseases such as cancer [38].

Calcium absorption during pregnancy increases up to 40% and supplementation is only recommended in pregnant women with an intake <600 mg/day, if there is a risk of preeclampsia, or in adolescent pregnancies [1]. With regard to this food source, overall, this study shows a good result since more than half of the pregnant women complied with the recommendations by including dairy products in their daily diets.

Regarding the consumption of olive oil, most pregnant women complied with the recommendations by consuming this product between 3–6 times a day, and this level was somewhat higher among those women with CV risk.

Although alcohol consumption during this period was minimal, as shown in Table 4, it should be noted that it was more abundant in the group of women who did not present CV risk.

### 3.4. Outcomes

The clinical outcomes of the pregnant women and their newborns as well as the biochemical values of the pregnant women participating in the study across the duration of their pregnancy are shown in Table 5 and Table 6, respectively. Among the clinical outcomes, a significant difference is observed with respect to the newborn classification for gestational age at birth. The group of women with CV risk presented a significantly higher risk of having small-for-gestational-age babies (RR: 1.20 (1.12–1.30) and RA: 16.9% (10.4–22.9)). Regarding the biochemical values assessed, women without CV risk presented better values of haematids and hemoglobin (*p* < 0.001) in the first two trimesters of pregnancy. The lack of difference in the third trimester could be due to supplementation in this period.

## 4. Discussion

This study has identified that there is a high prevalence of CV risk among the pregnant women studied at the La Fe Hospital in the Community of Valencia. It has been shown that choosing the MedDiet as a dietary pattern has numerous benefits, such as provoking an improvement in blood pressure and lipid levels, and a reduction in insulin resistance, adiposity, and dyslipidemia [12,13,14]. CV risk factors such as hypertension, type 2 diabetes mellitus, tobacco and alcohol consumption, dietary habits, physical exercise, and the relationship between apoprotein B-A1 can be modified based on one’s diet [40].

According to the National Survey of Spanish Dietary Intake (ENIDE) [41], the Spanish population currently has low adherence (62–75%) to the MedDiet, while high adherence to this pattern is only observed in 5–7%. In another study carried out in Spain during 2012 [42], adherence to this diet was evaluated using the Mediterranean Diet Adherence Screener (MEDAS) questionnaire and it was shown that 54% of the participants had strayed from this dietary pattern.

Regarding the adherence to this dietary pattern among pregnant women, a study carried out on the island of Menorca [43] maintains that 36.1% of the participating women showed low adherence to this pattern. In a study [44] carried out among a cohort of pregnant women from the South of Spain, it was shown that 40.2% do not follow a diet based on Mediterranean products. In a small study from Mérida (Spain), 37% of pregnant women presented low adherence to the MedDiet [45]. Another recent study from 2018 [46] indicates that 40.2% of mothers in the city of Valencia incorporate a low-quality MedDiet.

In this study, dietary patterns have been assessed using the MedDietScore questionnaire, which is not specific to the Spanish population. If we compare the dietary recommendations for Spanish pregnant women published by the SENC with the assumptions of the MedDiet made by the MedDietScore, we will observe some differences. In the categories of unrefined cereals, fruits, vegetables, and olive oil, both recommendations match; thus, the highest score in the MedDietScore will be achieved if the SENC recommendations are followed. In the case of whole dairy, the recommendations are contradictory: while the SENC recommends 3–4 servings per day, in the MedDietScore, whole dairy is one of the foods presumed to diverge from the Mediterranean pattern and the highest score is obtained when zero servings are consumed. In the case of the main protein sources, namely, legumes, fish, red meat, and chicken, the SENC recommendation is for two servings per day of protein-rich foods without specifying the source. Furthermore, in the MedDietScore, these protein sources are each scored differently, with legumes and fish being items presumed to be close to the Mediterranean pattern and red meat and chicken presumed to be foods away from the Mediterranean pattern.

There are certain factors that are associated with poor adherence to the MedDiet. A study conducted in Granada [47] found an inverse relationship between young age, economic and educational level, a sedentary lifestyle, and smoking and adherence to this pattern.

Obesity is the main risk factor to consider with respect to the development of maternal complications such as hypertension, gestational diabetes, preeclampsia, the risk of cesarean delivery, and other alterations in the fetus [48,49]. The MedDiet has also been associated with a lower average heart rate, the mitigation of the harmful effects of overweight/obesity on the risk of CV disease, and an attenuation of the effects of obesity on type 2 diabetes [50]. A study conducted in Germany [51] showed that having a high BMI prior to pregnancy significantly increased these alterations. Meanwhile, in a Swedish prospective study [10], obese mothers had a high probability of suffering from preeclampsia, caesarean section, and instrumented delivery in addition to other perinatal complications such as fetal distress or shoulder dystocia during vaginal delivery. In other non-European countries with dietary patterns different from Spanish gastronomy, it has been reported that an increase in weight and BMI before conception produces a greater predisposition to develop hypertensive disorders and gestational diabetes [52].

In this study, a higher pre-pregnancy weight was observed for the women with higher CV risk. Meanwhile, a study carried out at the Gran Canaria hospital [53] maintains that adherence to the Mediterranean diet before pregnancy has a protective effect on obesity and overweight, thus producing optimal weight gain during this period. This aspect may be due to the intake of plant foods rich in water, fiber, and other products with a low glycemic index. The increase in satiety, the release of cholecystokinin, and gastric distension are correlated with improved control of weight gain [54].

Adverse neonatal outcomes such as lower Apgar scores, hypoglycemia, seizures, meconium aspiration, increased weight for gestational age, infections, and prolonged hospital stay are associated with excess weight gain during pregnancy [55,56,57]. Adherence to the MedDiet was associated with lower odds of excessive gestational weight gain in a study conducted in the United Arab Emirates [58]. A multicenter, randomized trial study in the United Kingdom has shown that a simple, individualized, Mediterranean-style diet during pregnancy has the potential to reduce gestational weight gain and the risk of gestational diabetes [59]. Despite the existence of a high prevalence of the risk of coronary heart disease, most of the participants in our study complied with the recommendations for weight gain within the range of normal weight values. To achieve adequate weight gain during pregnancy, strategies based on a correct diet and regular physical exercise must be followed.

Another notable aspect of this study is the association between CV risk and the presence of gestational diabetes. A systematic review and meta-analysis published in 2021 concluded that women with previous gestational diabetes have a higher risk of developing CV disease based on significant increases in conventional risk factors. Some risk factors are seen as early as less than one year post-partum [60]. Risk factors that increase risk for future CV disease among women with a history of gestational diabetes include postpartum progression to type 2 diabetes mellitus; metabolic syndrome; obesity; hypertension; and altered levels of circulating inflammatory markers, specifically, adiponectin, C-reactive protein, and tumor necrosis factor-α [61].

Various studies have shown that dietary habits prior to pregnancy can influence the development of gestational diabetes [62], but few studies have investigated the consequences of different dietary patterns and gestational diabetes towards CV risk. A review by Amati et al. shows that gestational nutrition based on the MedDiet reduces the incidence of gestational diabetes [7]. Another study has shown that an early MedDiet-based nutritional intervention reduces the incidence of gestational diabetes and maternal–fetal adverse outcomes [63].

Zhang et al. [64] maintain that there is an increased risk of diabetes among women who follow a Western dietary pattern based on the consumption of red and processed meat, refined cereals, sweets, French fries, and pizza, unlike those who prefer a prudent diet characterized by the intake of fruits, vegetables, chicken, and fish [65]. On the contrary, a multicenter study carried out in ten Mediterranean countries shows that a high degree of adherence to the MedDiet is associated with a low incidence of gestational diabetes [66]. Note that following a dietary pattern characterized by processed foods (salt, sugary drinks, snacks, and processed meats) has also been associated with a higher probability of developing hypertensive disorders (preeclampsia and hypertension) during pregnancy. No effect was observed between the consumption of products rich in simple carbohydrates (sweets and cakes) and the risk of preeclampsia [67].

However, in this study, the quality of the diet is analyzed in a generic way, and, therefore, the exact amounts of nutrients that the participants consumed during pregnancy are unknown. This aspect is important since previous studies have shown that certain isolated nutrients contribute to increasing CV risk [2,4].

Several studies have shown that consuming a low amount of fiber and foods with a high glycemic index influences the development of gestational diabetes [68]. Nutritional interventions carried out in pregnant women support the notion that a diet high in complex carbohydrates and fiber and a reduction in the consumption of simple sugar and saturated fat produces an improvement in glycemia and prevents insulin resistance [69]. However, another prospective study [65] found no relationship between the quality of carbohydrates and the different types of fatty acids and diabetes.

Smoking during pregnancy implies a high nutritional risk for the mother and directly affects the development of the fetus due to a decrease in the absorption of nutrients through the placenta. In this group, it is common to eat fewer foods rich in antioxidants; consequently, this leads to low dietary quality that affects the assimilation of certain micronutrients [70].

The results obtained conclude that smoking mothers have a higher CV risk compared to the non-smoker profile. A study in England [71] verified this association by revealing that pregnant women who smoke have an unhealthy diet in addition to being overweight at the beginning of pregnancy. Smoking has been correlated with women who opt for a Western dietary pattern, weakly adhere to the MedDiet, and hardly engage in physical activity [46,72,73].

In this study, although approximately 80% of the participants perform physical activity, generally, light activity predominates, with moderate activity being higher in women with adequate adherence to the MedDiet.

A sedentary lifestyle is a conditioning factor in diets, and it has been shown that physical inactivity is characteristic and superior in those women whose diet is insufficiently healthy [74].

After a systematic review [75], it can be concluded that only interventions in physical exercise can reduce the probability of developing gestational diabetes by 38%, gestational hypertension by 39%, and preeclampsia by 41%.

Considering the state and characteristics of the pregnancy, the WHO [76] recommends performing at least 150 min of moderate physical activity per week, 75 min of intense activity, or a combination of both types of activity. Some benefits associated with performing moderate physical exercise among healthy, pregnant women include a good cardiorespiratory condition, fewer symptoms of postpartum depression, and a low probability of maintaining the weight gained after the pregnancy period [77].

Certain micronutrients must be supplemented before and during pregnancy since most women often do not meet the recommended requirements [29]. Supplementation with 400 μg of folic acid is recommended 1 month before conception for at least the first 2–3 months of pregnancy and in risk situations during the nine months [1]. In this study, the results were positive since there was higher consumption of folic acid by women with CV risk. On the contrary, a study carried out in the Community of Valencia [44] shows that folic acid intake was lower in pregnant women who had low adherence to the MedDiet. This vitamin B9 deficiency can cause megaloblastic anemia in women, neural tube defects, and birth defects in children [78].

Iron supplementation (82%) was high in both groups of women, with hardly any differences between them. There are contradictions regarding the supplementation of this mineral, since its use is suggested in women with iron deficiency or iron deficiency-related anemia, but it is routinely limited in women without risk [1].

Elevated iron levels are associated with gastrointestinal disorders, increased oxidative stress, and the production of free radicals [79]. In another study carried out in the Mediterranean area [80], it was observed that almost none of the pregnant women met the recommendations for folic acid consumption and 68% did not reach those for iron either.

Fish was the least consumed food by pregnant women; thus, their main sources of protein are obtained through the intake of meat and dairy products. Two studies carried out in Mediterranean areas maintain that 56–69% of women include fish in their diet less than twice a week [46,81]. Omega-3 essential fatty acids play a fundamental role in the cognitive, neurological, and visual development of the fetus and in CV health during adulthood [78]. The main sources of this type of acid are fish, shellfish, and fish oils [82,83]. The consumption of fatty fish (salmon, herring, and mackerel) one to two times per week or marine oils would cover the recommended intake for Spanish women that stipulates the consumption of at least 200 mg/day of DHA [84].

It has been proven that supplementation with omega-3 fatty acids, marine oils, and other prostaglandin precursors leads to a lower probability of preterm birth and increased weight and size of the fetus, although it probably increases the incidence of post-term pregnancies [85]. Despite the positive effects obtained, due to the possibility for a reduced risk of perinatal death and of neonatal care admission, a reduced risk of low-birth-weight babies, and a slightly increased risk of large-for-gestational-age babies via omega-3 supplementation, there is not enough evidence to advise the supplementation of marine oils during pregnancy, as further follow-ups of the completed trials are needed in order to assess longer-term outcomes [85].

Vitamins A, D, E, and B6 are considered important as the requirements are increased during pregnancy and intervene in fetal growth and in the development of the placenta [1]. However, more evidence is needed to indicate the regular supplementation of these micronutrients since, so far, it is only recommended for women who present deficiencies.

Calcium and iodine are two essential minerals throughout pregnancy [4]. The general recommendation established by some health organizations indicates taking 150–200 μg/day of iodine during pregnancy [29].

## 5. Strengths and Limitations

This study has limitations. First, the sample size is not very large; therefore, there is a greater lack of control of the confounding factors, so it is advisable to increase the number of participants to obtain more solid and reliable results.

In addition, the collection of information through the MedDietScore questionnaire was only performed once, so there is a higher risk of recall bias. The results obtained from these interviews were reported to a program called MedDietScore to estimate adherence to the MedDiet and CV risk. It must be considered that the score ranges of the questionnaire given to the respondents and the values of the program used were not the same; therefore, the data have been interpreted in such a way that there may be a small margin of error.

This program is an effective tool to reduce and prevent food-related diseases. It is an index used by the adult population and, to date, no other specific index is known for special groups such as pregnant women [34]. Other studies have been carried out on pregnant women excluding alcohol consumption from the different indices since it is normally absent and not recommended [66,70].

## 6. Conclusions

In conclusion, 54.2% of the women considered without CV risk and 45.8% of the women with CV risk obtained a score greater than 35 for the evaluation of the diet; therefore, it can be argued that they maintain a good degree of adherence to the MedDiet according to the MedDietScore.

When considering the recommendations indicated by the SENC, the main food groups in which recommendations are not met are unrefined cereals and vegetables. Meanwhile, regarding the intake of fruits and dairy products, 65% and 55% of the participants met the recommendations. As for protein-rich foods, the main sources of protein were red meat and chicken, with fish being the least common.

Concerning the risk of CV, the promotion and encouragement of strategies based on a correct diet and the practice of physical exercise are essential to avoid the development of possible chronic diseases that can affect both the mother and the newborn in later stages.

## Figures and Tables

**Figure 1 life-13-00241-f001:**
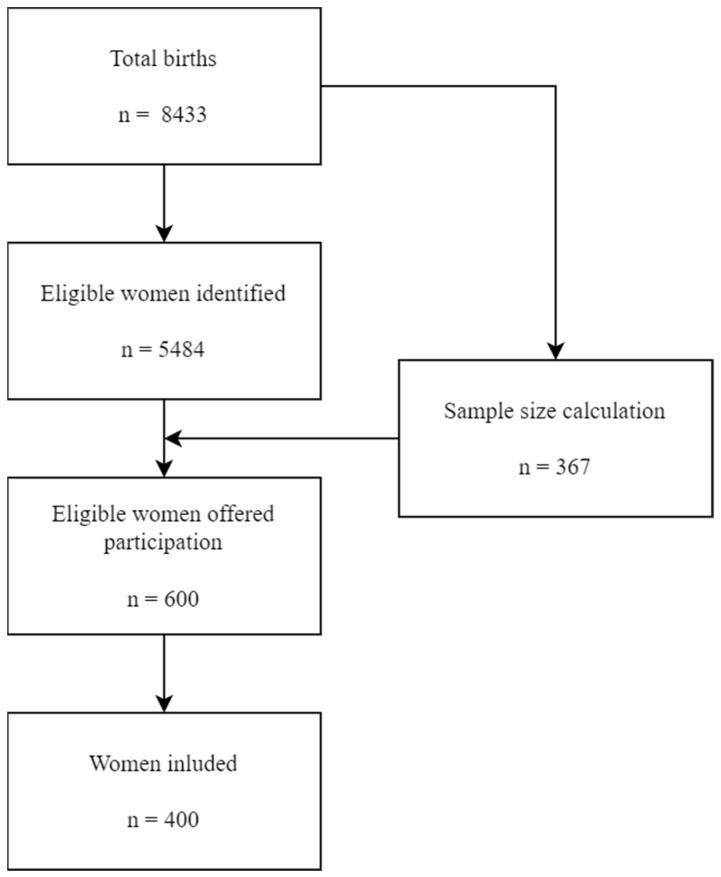
Participant recruitment.

**Figure 2 life-13-00241-f002:**
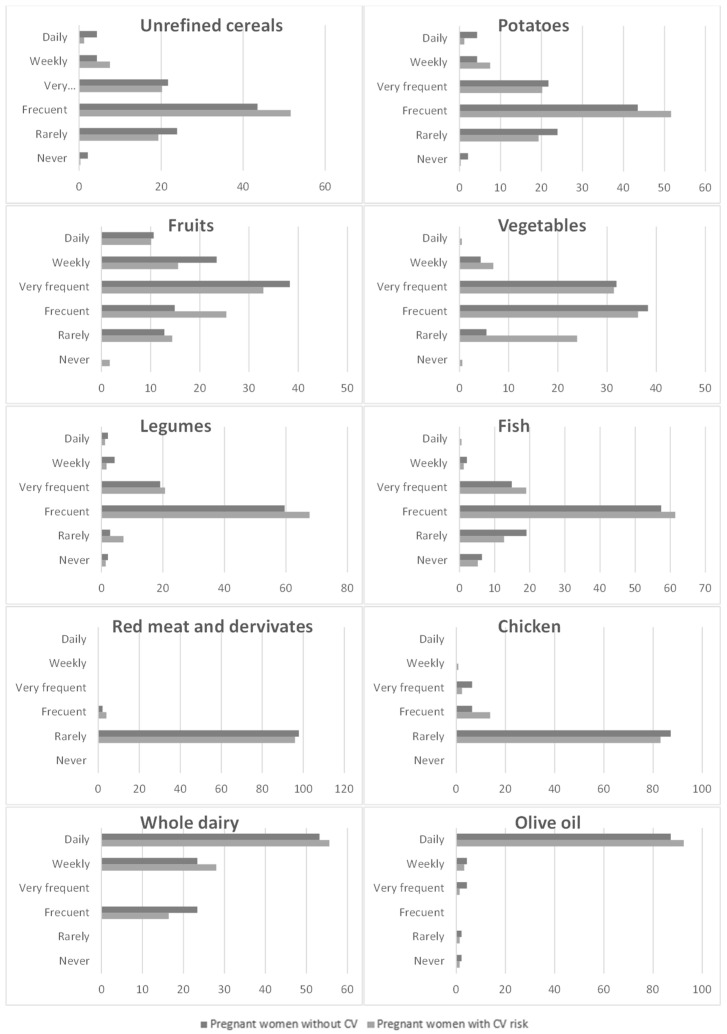
Food consumption corresponding to the Mediterranean diet pattern.

**Table 1 life-13-00241-t001:** Personal characteristics and habits of the pregnant women participating in the study.

	Women with Cardiovascular Risk (n = 349) % (95%CI)/Mean ± SD	Women without Cardiovascular Risk (n = 51) % (95%CI)/Mean ± SD	*p-Value* *
n (%)	87.25 [83.48–90.27]	12.75 [9.72–16.51]	-
Age (years)	32.38 ± 5.73	33.31 ± 5.55	0.291
Pre-pregnancy weight (Kg) Final weight (Kg) Weight gained (Kg)	65.00 ± 12.67 76.89 ± 13.12 12.00 ± 8.80	63.99 ± 12.06 76.73 ± 12.78 12.74 ± 5.13	0.600 0.935 0.571
BMI before pregnancy BMI 3rd trimester	24.25 ± 5.10 28.77 ± 5.16	24.25 ± 4.45 29.05 ± 4.56	0.994 0.712
Height (m)	1.63 ± 0.06	1.62 ± 0.06	0.513
Nulliparous (%)	51.00 [45.63–56.64]	50.00 [35.42–64.57]	0.997
Race/Ethnicity European African American Asian	84.00 [79.58–87.56] 4.60 [2.73–7.48] 9.40 [6.69–13.14] 2.00 [0.88–4.27]	91.70 [79.12–97.29] 6.30 [1.62–18.20] 2.00 [0.10–12.47] 0.00 [0.00–9.23]	0.161 0.883 0.150 -
Gestation weeks	39.25 ± 1.93	38.70 ± 2.21	0.072
Gestational diabetes (%)	9.73 [6.84–13.58]	6.97 [1.81–20.12]	0.606
Education No formal education Primary Secondary Tertiary	4.29 [2.42–6.98] 14.32 [10.82–18.44] 45.27 [39.98–50.66] 36.10 [31.05–41.38]	1.96 [0.10–11.79] 11.76 [4.44–23.86] 29.41 [17.48–43.82] 50.98 [36.77–65.04]	0.679 0.622 0.032 0.040
Employment status Employed Unemployed	65.61 [60.37–70.59] 34.38 [29.40–39.62]	64.70 [50.06–77.56] 27.45 [15.89–41.74]	0.898 0.327
Place of residence Urban (Population > 10,000) Rural (Population < 10,000)	100.0 [100.0–100.0] 0.0 [0.00–0.00]	100.0 [100.0–100.0] 0.0 [0.00–0.00]	- -
Physical activity (%) None Light Moderate	77.10 [72.23–81.31] 22.90 [18.24–28.63] 66.05 [59.37–71.03] 11.05 [7.14–14.49]	79.20 [66.45–89.71] 20.80 [11.11–38.03] 62.50 [46.91–77.43] 16.70 [7.69–32.64]	0.502 0.817 0.717 0.321
Tobacco use in 1st trimester (%)	13.18 [9.90–17.29]	10.41 [3.89–23.44]	0.004
Alcohol consumption (%)	2.50 [1.26–5.02]	6.25 [1.62–18.20]	0.165
Folic acid intake (%)	71.34 [66.24–75.97]	54.16 [39.30–68.36]	0.202
Iron intake (%)	82.70 [78.22–86.45]	81.25 [66.89–90.56]	0.803
Vitamin supplements (%)	87.57 [83.51–90.76]	80.85 [66.27–90.35]	0.202
Food supplements (%)	0.28 [0.015–1.84]	0.00 [0.00–9.23]	0.710

* *p-Value* obtained from the Chi Test for qualitative variables and ANOVA test for quantitative variables.

**Table 2 life-13-00241-t002:** Intake frequency according to the Mediterranean diet pattern score (MedDietScore) of the pregnant women participating in the study.

	Never (0 Times/Month) % [95%CI]	Rarely (1–4 Times/Month) % [95%CI]	Frequent (5–8 Times/Month) % [95%CI]	Very frequent (9–12 Times/Month) % [95%CI]	Weekly (13–18 Times/Month) % [95%CI]	Daily (>18 Times/Month) % [95%CI]	*p-Value* *
Women with cardiovascular risk (n = 349)							
Unrefined cereals	0.3 [0.15–1.83]	19.3 [15.27–23.80]	51.6 [45.91–56.63]	20.2 [16.06–24.72]	7.5 [5.01–10.85]	1.2 [0.36–3.11]	<0.001
Potatoes	0.9 [0.22–2.70]	67.1 [61.22–71.36]	24.9 [20.28–29.57]	4.9 [2.95–7.83]	1.7 [0.70–3.89]	0.6 [0.09–2.28]	<0.001
Fruits	1.7 [0.70–3.89]	14.4 [10.91–18.54]	25.4 [20.81–30.17]	32.9 [27.82–37.89]	15.6 [1.90–19.79]	10.1 [7.17–13.78]	<0.001
Vegetables	0.6 [0.09–2.28]	23.9 [19.48–29.67]	36.3 [31.10–41.41]	31.4 [26.46–36.42]	6.9 [4.78–10.52]	0.5 [0.09–2.28]	<0.001
Legumes	1.4 [0.52–3.50]	7.2 [4.78–10.52]	67.7 [62.10–72.17]	20.7 [16.58–25.33]	1.7 [0.70–3.89]	1.2 [0.36–3.11]	<0.001
Fish	5.2 [3.17–8.17]	12.7 [9.40–16.66]	61.4 [55.67–66.13]	19.0 [15.01–23.5]	1.2 [0.36–3.11]	0.6 [0.09–2.28]	<0.001
Red meat and derivatives	0.0 [0.00–1.35]	96.0 [92.51–87.26]	4.0 [2.29–6.79]	0.0 [0.00–1.35]	0.0 [0.00–1.35]	0.0 [0.00–1.35]	<0.001
Chicken	0.0 [0.00–1.35]	83.0 [78.03–86.27]	13.8 [10.40–17.92]	2.3 [1.06–4.64]	0.9 [0.22–2.70]	0.0 [0.00–1.35]	<0.001
Whole dairy	0.0 [0.00–1.35]	0.0 [0.00–1.35]	16.4 [12.69–20.72]	0.0 [0.00–1.35]	28.0 [23.22–32.86]	55.6 [49.91–60.57]	<0.001
Use of olive oil	1.4 [0.52–3.50]	1.4 [0.52–3.50]	0.0 [0.00–1.35]	1.4 [0.52–3.50]	3.2 [1.66–5.73]	92.5 [88.48–94.51]	<0.001
Alcoholic drinks	100.0 [100.0–100.0]	0.0 [0.00–0.00]	0.0 [0.00–0.00]	0.0 [0.00–0.00]	0.0 [0.00–0.00]	0.0 [0.00–0.00]	-
Women without cardiovascular risk (n = 51)							
Unrefined cereals	2.1 [0.10–12.47]	23.9 [12.51–37.66]	43.5 [27.93–56.72]	21.7 [10.95–35.40]	4.3 [0.72–15.42]	4.3 [0.72–15.42]	<0.001
Potatoes	2.1 [0.10–12.47]	57.4 [41.28–70.22]	34.0 [20.80–48.51]	4.3 [0.72–15.42]	2.1 [0.10–12.47]	0.0 [0.00–9.23]	<0.001
Fruits	0.0 [0.00–9.23]	12.8 [5.18–25.94]	14.9 [6.54–28.37]	38.3 [24.32–52.66]	23.4 [12.51–37.66]	10.6 [3.89–23.44]	<0.001
Vegetables	0.0 [0.00–9.23]	5.5 [14.10–39.89]	38.3 [24.32–52.66]	31.9 [9.09–46.40]	4.3 [0.72–15.42]	0.0 [0.00–9.23]	<0.001
Legumes	2.1 [0.10–12.47]	2.8 [5.18–25.94]	59.6 [43.27–72.06]	19.1 [9.43–33.10]	4.3 [0.72–15.42]	2.1 [0.10–12.47]	<0.001
Fish	6.4 [1.62–18.20]	19.1 [9.43–33.10]	57.4 [41.70.22]	14.9 [6.54–28.37]	2.1 [0.10–12.47]	0.0 [0.00–9.23]	<0.001
Red meat and derivatives	0.0 [0.00–9.23]	97.9 [84.57–99.27]	2.1 [0.10–12.47]	0.0 [0.00–9.23]	0.0 [0.00–9.23]	0.0 [0.00–9.23]	<0.001
Chicken	0.0 [0.00–9.23]	87.2 [71.62–93.45]	6.4 [1.62–18.20]	6.4 [1.62–18.20]	0.0 [0.00–9.23]	0.0 [0.00–9.23]	<0.001
Whole dairy	0.0 [0.00–9.23]	0.0 [0.00–9.23]	23.4 [12.51–37.66]	0.0 [0.00–1.35]	23.4 [12.51–37.66]	53.2 [37.35–66.47]	<0.001
Use of olive oil	2.1 [0.10–12.47]	2.1 [0.10–12.47]	0.0 [0.00–9.23]	4.3 [0.72–15.42]	4.3 [0.72––15.42]	87.2 [71.62–93.45]	<0.001
Alcoholic drinks	100.0 [100.0–100.0]	0.0 [0.00–0.00]	0.0 [0.00–0.00]	0.0 [0.00–0.00]	0.0 [0.00–0.00]	0.0 [0.00–0.00]	-
Comparative intake ratio (Women with cardiovascular risk/Women without cardiovascular risk)							
Unrefined cereals	0.14	0.80	1.18	0.93	1.74	0.27	-
Potatoes	0.42	1.16	0.73	1.13	0.80	0.60	-
Fruits	1.70	1.12	1.70	0.85	0.66	0.95	-
Vegetables	0.60	4.34	0.95	0.98	1.60	0.50	-
Legumes	0.66	2.57	1.13	1.08	0.39	0.57	-
Fish	0.81	0.66	1.07	1.27	0.57	0.60	-
Red meat and derivatives	1.00	0.98	1.90	1.00	1.00	1.00	-
Chicken	1.00	0.95	2.15	0.36	0.90	1.00	-
Whole dairy	1.00	1.00	0.70	1.00	1.19	1.04	-
Use of olive oil	0.67	0.67	1.00	0.32	0.74	1.06	-
Alcoholic drinks	1.00	1.00	1.00	1.00	1.00	1.00	-

* *p-Value* obtained from the Chi Test for qualitative variables and ANOVA test for quantitative variables.

**Table 3 life-13-00241-t003:** Compliance with Spanish Society of Community Nutrition (SENC) dietary recommendations of the pregnant women participating in the study [37].

	SENC Recommendation (Servings/Day)	Compliance		
		Women with Cardiovascular Risk (n = 349) % [95%CI]	Women without Cardiovascular Risk (n = 51) % [95%CI]	*p-Value* *
Unrefined cereals	4–5	1.2 [0.36–3.11]	4.3 [0.72–15.42]	-
Fruits	2–3	15.6 [1.9–19.79]	23.4 [12.51–37.66]	0.147
Vegetables	2–4	6.9 [4.78–10.52]	4.3 [0.72–15.42]	0.620
Legumes	2 **	23.6 [19.21–28.36]	25.5 [14.77–39.91]	0.754
Fish	2 **	20.8 [16.58–25.33]	17.0 [8.86–31.36]	0.588
Red meat and derivatives	2 **	0	0	-
Chicken	2 **	3.16 [26.73–36.71]	6.4 [1.53–17.22]	0.559
Whole dairy	3–4	55.6 [49.91–60.57]	53.2 [37.35–66.47]	0.722
Olive oil	3–6	92.5 [88.48–94.51]	87.2 [71.62–93.45]	0.211

* *p-Value* obtained from the Chi Test for qualitative variables and ANOVA test for quantitative variables. ** SENC recommendation is for 2 servings per day of protein-rich foods.

**Table 4 life-13-00241-t004:** Beverage intake of the pregnant women participating in the study.

	Women with Cardiovascular Risk (n = 349) n/% [95%CI]	Women without Cardiovascular Risk (n = 51) % [95%CI]	*p-Value **
Alcohol	9/2.57 [1.26–5.01]	3/5.88 [1.53–17.22]	0.927
Beer	8/2.29 [1.06–4.64]	3/5.88 [1.53–17.22]	0.291
Wine	3/0.86 [0.22–2.70]	2/3.92 [0.68–14.59]	0.600
Spirits	1/0.28 [0.01–1.83]	0	0.994
Soft drinks	130/37.24 [32.20–42.58]	22/43.13 [29.62–57.14]	0.513
Coffee	161/46.13 [40.83–51.51]	23/45.09 [31.38–59.54]	0.612

* *p-Value* obtained from the Chi Test for qualitative variables and ANOVA test for quantitative variables.

**Table 5 life-13-00241-t005:** Clinical outcomes of the pregnant women participating in the study and their newborns.

	Women with Cardiovascular Risk (n = 349) n/% [95%CI]	Women without Cardiovascular Risk (n = 51) % [95%CI]	*p **	*Relative Risk*	*Attributable Risk*
Preeclampsia	9 (2.58 (1.26–5.01)	2 (3.92 (0.68–14.59)	0.928	0.94 (0.71–1.24)	-
Neonatal ICU admission	2 (0.57 (0.10–2.28)	0	-	-	-
Newborn classification for gestational age					
Normal	167 (47.52 (42.52–53.22)	43 (84.31 (70.86–92.51)	0.001	1	-
Small	176 (50.42 (45.06–55.79)	8 (15.68 (7.48–29.13)	0.003	1.20 (1.12–1.30)	16.9% (10.4–22.9)
Large	6 (1.71 (0.69–3.89)	0	-	1.08 (0.79–1.47)	7.2% (−26.5–32.0)

** p-Value* obtained from the Chi Test for qualitative variables and ANOVA test for quantitative variables.

**Table 6 life-13-00241-t006:** Biochemical values of the pregnant women participating in the study across the duration of their pregnancy.

	Women with Cardiovascular Risk (n = 349) Mean ± SD	Women without Cardiovascular Risk (n = 51) Mean ± SD	*p-Value* *
O’Sullivan test	129.01 ± 30.12	139.75 ± 24.76	0.321
Glucose (1st trimester)	83.57 ± 8.28	93.50 ± 14.85	0.113
Haematids (1st trimester)	5.33 ± 2.50	9.08 ± 3.47	<0.001
Hemoglobin (1st trimester)	13.86 ± 7.93	12.57 ± 1.47	0.779
Glucose (2nd trimester)	68.97 ± 17.78	72.00 ± 12.40	0.868
Haematids (2nd trimester)	8.02 ± 4.07	11.18 ± 2.70	<0.001
Hemoglobin (2nd trimester)	22.78 ± 12.15	31.95 ± 10.50	<0.001
Glucose (3rd trimester)	72.48 ± 13.45	74.00 ± 9.90	0.876
Haematids (3rd trimester)	3.91 ± 0.31	4.02 ± 0.58	0.451
Hemoglobin (3rd trimester)	11.66 ± 1.02	11.03 ± 1.61	0.155
Hematocrit (3rd trimester)	34.96 ± 2.74	33.40 ± 4.8	0.188
Iron (3rd trimester)	79.42 ± 48.11	44.50 ± 30.41	0.31
Ferritin (3rd trimester)	21.94 ± 26.73	6.50 ± 9.19	0.418
Hemoglobin (Birth)	12.03 ± 1.59	11.78 ± 1.32	0.659
Hematocrit (Birth)	35.68 ± 4.34	35.20 ± 2.97	0.757

* *p-Value* obtained from the Chi Test for qualitative variables and ANOVA test for quantitative variables.

## Data Availability

Not applicable.

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
