# Peer review of "Mediterranean Dietary Pattern and Cardiovascular Risk in Pregnant Women"

_life, 2023, doi:10.3390/life13010241_

Round 1
Reviewer 1 Report
Review of Life-2076761 Title: Mediterranean Dietary Pattern and Cardiovascular Risk in Pregnant Women The authors undertook a an interesting topic about the relationship between Mediterranean dietary pattern and cardiovascular risk in pregnant women. The Abstract section – it's written correctly, however, Simple Summary is redundant and not required in the Instruction for Authors. The Introduction section - The content of the Introduction section is not consistent. In verses 53-56, the Authors indicate that epidemiological studies on the preventive role of the Mediterranean diet in the context of cardiovascular diseases, give effects with long-term use and pregnancy lasts "only" nine months. On the other hand, the Authors indicate nutrients whose excess or deficiency may potentially affect the risk of pregnancy complications, although as mentioned earlier, the intake of individual nutrients does not reflect dietary patterns. Therefore, there is no link between the MedDiet nutritional pattern and the nutrients listed by the Authors. The Methods section - The recruitment process for the study would be clearer if it were presented in the form of a flowchart diagram. In the Personal characteristics section, there is no information in which week of pregnancy the study participants were recruited. The criteria for inclusion and exclusion from the study were also not specified. It is puzzling why no questions were asked about education, economic status and place of residence - these are the main factors determining food choices. In verse 143, the authors provide "Maternal physical activity and employment status during the months of pregnancy were assessed" - it is not known what method was used to assess activity or to what period of time the question about physical activity referred to - the period before pregnancy or only during pregnancy - it is extremely difficult assess the status of physical activity, for example, in a pregnant woman with a high-risk pregnancy and a ban on any activity, who had high physical activity before pregnancy. There is no reference to the employment of pregnant women in any of the tables or in the subsequent text. In the Dietary assessment section, the period covered by the questionnaire was not specified - only the period of pregnancy or the last year, was the questionnaire completed only once (in which trimester?) or once in each trimester? Was the questionnaire used validated in the group of pregnant women? - if not then it should not be used in this group. All the questions asked indicate the possibility of making several serious methodological errors. The Results sections - The authors repeated the same text in the Cardiovascular risk section (lines 188-191) and in the Food consumption of the Mediterraneandet pattern section (lines 206-208) and in the Outcomes section (lines 277-281) - in the last two cases as well as in verse 281 the sentences seem to to be unfinished. It is puzzling why Table 2 does not include the results for alcoholic drinks when their consumption is shown in Table 1 and Table 4. The Discussion section - In verse 301, the authors refer to the MEDAS questionnaire - the skin should be explained. In the text of the Discussion section, the authors refer to studies on the analysis of pre-pregnancy eating behaviors, and in the presented study, the analysis was probably conducted only during pregnancy - these are incomparable periods. What is missing from this section is the comparison of dietary recommendations for Spanish pregnant women with the assumptions of the Mediterranean diet. The Conclusions section - it is surprising why the Authors in the first sentence refer to the recommendations for pregnant women developed by the Spanish Society of Community Nutrition and not the MedDiet pattern. All of the above comments indicate that the manuscript in its present form is unsuitable for publication..
Author Response
The authors undertook an interesting topic about the relationship between Mediterranean dietary pattern and cardiovascular risk in pregnant women.
We greatly appreciate the time and effort the reviewer has taken to help us improve our work. We hope to have addressed all their concerns accordingly but also remain open to receiving further feedback and making any additional clarifications or changes deemed necessary.
Note: In the marked version of the manuscript the references are not corrected as we have used the Track Changes function which makes using the Citation Manager in Word a little messy. We have, however, provided a clean version with the corrected references.
The Abstract section – it's written correctly, however, Simple Summary is redundant and not required in the Instruction for Authors.
We are not sure where the Simple Summary came from as we did not include it in the manuscript that we uploaded. We have found other numerous problems with the formatting changes made by the journal so perhaps there might be an issue there…
The Introduction section - The content of the Introduction section is not consistent. In verses 53-56, the Authors indicate that epidemiological studies on the preventive role of the Mediterranean diet in the context of cardiovascular diseases, give effects with long-term use and pregnancy lasts "only" nine months. On the other hand, the Authors indicate nutrients whose excess or deficiency may potentially affect the risk of pregnancy complications, although as mentioned earlier, the intake of individual nutrients does not reflect dietary patterns. Therefore, there is no link between the MedDiet nutritional pattern and the nutrients listed by the Authors.
We have revised this whole section and a great part of it has been rewritten taking the reviewer’s comments into account. We understand why the reviewer points out that effects of the MedDiet are seen after long-term use and pregnancy is a very limited period of time. However, it is a time in which most women will take special care of their nutrition, even if food aversions, nausea and other pregnancy symptoms sometimes affect food intake. The dietary pattern followed during this time should be reflective of their habitual long-term eating habits and may be even somewhat better, therefore, we feel that it is suitable for analysis in the context of the associations between MedDiet and its effects on CV health.
The Methods section - The recruitment process for the study would be clearer if it were presented in the form of a flowchart diagram.
We have included a clearer and more detailed explanation of the recruitment process but do not think that a flowchart diagram would add significant value. We would, however, include it if the reviewer deems it necessary.
In the Personal characteristics section, there is no information in which week of pregnancy the study participants were recruited.
Women were offered participation in this specific study during a post-birth visit but have been followed using the same care protocol since pregnancy confirmation before 12 weeks.
The criteria for inclusion and exclusion from the study were also not specified.
The inclusion and exclusion criteria have been included in the previous section.
It is puzzling why no questions were asked about education, economic status and place of residence - these are the main factors determining food choices.
This information was collected but since it is not presented, it was not mentioned. We could add this information to Table 1 if the reviewer thinks it would add to our work.
In verse 143, the authors provide "Maternal physical activity and employment status during the months of pregnancy were assessed" - it is not known what method was used to assess activity or to what period of time the question about physical activity referred to - the period before pregnancy or only during pregnancy - it is extremely difficult assess the status of physical activity, for example, in a pregnant woman with a high-risk pregnancy and a ban on any activity, who had high physical activity before pregnancy. There is no reference to the employment of pregnant women in any of the tables or in the subsequent text.
Physical activity during pregnancy was classified into the categories none, light and moderate using the self-reported information provided by the women. We understand that it is extremely difficult assess the status of physical activity especially given all the different situations the women may be in, however, as our aim is to assess CV risk due to diet it should not impact our results even if it would obviously have an impact on overall CV risk.
It is true that we state that information on employment status during pregnancy was collected and then not presented. We could add this information, as well as that about education, economic status, and place of residence to Table 1 if the reviewer thinks it would add to our work.
In the Dietary assessment section, the period covered by the questionnaire was not specified - only the period of pregnancy or the last year, was the questionnaire completed only once (in which trimester?) or once in each trimester? Was the questionnaire used validated in the group of pregnant women? - if not then it should not be used in this group. All the questions asked indicate the possibility of making several serious methodological errors.
Women were instructed to complete the validated MedDietScore questionnaire once, during their last programed antenatal visit (36-38 weeks), to reflect their habitual diet during the whole of their pregnancy.
The Results sections - The authors repeated the same text in the Cardiovascular risk section (lines 188-191) and in the Food consumption of the Mediterranean diet pattern section (lines 206-208) and in the Outcomes section (lines 277-281) - in the last two cases as well as in verse 281 the sentences seem to be unfinished.
We have corrected this issue in the text, once again, we think this was an issue with the formatting.
It is puzzling why Table 2 does not include the results for alcoholic drinks when their consumption is shown in Table 1 and Table 4.
Table 2 is a reflection of habitual diet while Table 1 and Table 4 reflects the answers to the YES/NO question of if the women consumed alcohol at any point during pregnancy. All women answered Never (0 times/month) in the MedDietScore questionnaire, but some did say that at some point they had consumed alcohol. This could be for example, just one beer during all pregnancy. Given the answer to the MedDiet, question, that is why it was not presented but hawe have included it now.
The Discussion section - In verse 301, the authors refer to the MEDAS questionnaire - the skin should be explained.
MEDAS refers to the Mediterranean Diet Adherence Screener.
In the text of the Discussion section, the authors refer to studies on the analysis of pre-pregnancy eating behaviors, and in the presented study, the analysis was probably conducted only during pregnancy - these are incomparable periods.
We included information on the eating behaviors of the general adult Spanish population in order to provide context for the current situation regarding dietary behaviors in Spain but will removed it if the reviewer does not think it adds relevant information.
What is missing from this section is the comparison of dietary recommendations for Spanish pregnant women with the assumptions of the Mediterranean diet.
We have included a comparison among the two in the Discussion as suggested.
The Conclusions section - it is surprising why the Authors in the first sentence refer to the recommendations for pregnant women developed by the Spanish Society of Community Nutrition and not the MedDiet pattern.
We have rewritten the Discussion in order to better reflect the objective and results of this work.
All of the above comments indicate that the manuscript in its present form is unsuitable for publication.

Reviewer 2 Report
The authors present a population-based cross-sectional study which aims to investigate the relationship between keeping Mediterranean diet and cardiovascular risk in pregnant women. Although the study is properly conducted and the manuscript is well written, several corrections should be made. First, the introduction part needs shortening. Second, the study is a cross-sectional, and not a prospective one. Therefore, the word "prospective" should be omitted from the third sentence of the materials & methods part. Third, the references that were published before 2015 should be replaced with newer and more up-to-date ones if possible.
Author Response
The authors present a population-based cross-sectional study which aims to investigate the relationship between keeping Mediterranean diet and cardiovascular risk in pregnant women.
Although the study is properly conducted and the manuscript is well written, several corrections should be made.
We greatly appreciate the time and effort the reviewer has taken to help us improve our work. We hope to have addressed all their concerns accordingly but also remain open to receiving further feedback and making any additional clarifications or changes deemed necessary.
Note: In the marked version of the manuscript the references are not corrected as we have used the Track Changes function which makes using the Citation Manager in Word a little messy. We have, however, provided a clean version with the corrected references.
First, the introduction part needs shortening.
We have rewritten a substantial part and shortened the Introduction as requested.
Second, the study is a cross-sectional, and not a prospective one. Therefore, the word "prospective" should be omitted from the third sentence of the materials & methods part.
We have made this correction in the text as required.
Third, the references that were published before 2015 should be replaced with newer and more up-to-date ones if possible.
References published before 2015 have been replaced with newer more up to date ones when possible.

Reviewer 3 Report
The manuscript entitled “ Mediterranean Dietary Pattern and Cardiovascular Risk in Pregnant Women" The manuscript is well written. It introduces a good knowledge and scientific soundness. I found the authors worked well on their paper. Some minor concerns are needed to be modified for improvement of the manuscript.
- In the abstract :” Our study identified that 87.25% (95%CI: 83.48-90.27) 33 of the subjects had cardiovascular risk in relation with their dietary intake” I would like the authors to add some examples for the types of food they mostly consume.
- “ weighed more at the beginning of pregnancy, performed little physical activity, 35 and had less adherence to the MedDiet compared to women who did not suffer from said risk.” These need rephrasing as they are somewhat confusing.
- I suggest the authors replace the keywords with other than that of the manuscript title for better relevance to wide range of searches.
- Line 48 : needs rephrasing “ not clear”
- Line 65 “ are included in a pattern dietary” correct. I think it is wrong presented
- Line 104 “ ´A study carried out in China in 2015 “ I think this should be followed by past sentence “ showed”
- I can’t understand how could you estimate that the CV is related to the diet factors not due to any other causes including pregnancy “ MedDiet and estimate CV risk based on diet”
- How could you synchronize the risk of CV with the stage of the pregnancy in the tested groups especially who are at high risk?
- Also the presence of different races in another areas they got accustomed, may be other factor that should be taken in consideration
- The footnote of the table should precisely describe all abbreviations in the table
- Line 283 : correct “ val-ues”
- The P value should be written in the footnote.
- I think lines from 312- 315 need revision , I can’t get clear point.
- The association between cardiovascular risk and the presence of gestational diabetes, is not well addressed in the discussion and needs further recent citations.
Author Response
The manuscript entitled “Mediterranean Dietary Pattern and Cardiovascular Risk in Pregnant Women" The manuscript is well written. It introduces a good knowledge and scientific soundness. I found the authors worked well on their paper. Some minor concerns are needed to be modified for improvement of the manuscript.
We greatly appreciate the time and effort the reviewer has taken to help us improve our work. We hope to have addressed all their concerns accordingly but also remain open to receiving further feedback and making any additional clarifications or changes deemed necessary.
Note: In the marked version of the manuscript the references are not corrected as we have used the Track Changes function which makes using the Citation Manager in Word a little messy. We have, however, provided a clean version with the corrected references.
- In the abstract:” Our study identified that 87.25% (95%CI: 83.48-90.27) 33 of the subjects had cardiovascular risk in relation with their dietary intake” I would like the authors to add some examples for the types of food they mostly consume.
According to their answers in the MedDietScore questionnaire, the high CV risk in these 33 patients is mainly due to not consuming certain types of foods, namely fruits, vegetables and legumes, more so then consuming foods that elevate their risk.
- “weighed more at the beginning of pregnancy, performed little physical activity, 35 and had less adherence to the MedDiet compared to women who did not suffer from said risk.” These need rephrasing as they are somewhat confusing.
We have rewritten this sentence as suggested.
- I suggest the authors replace the keywords with other than that of the manuscript title for better relevance to wide range of searches.
Change has been made as suggested.
- Line 48: needs rephrasing “not clear”
Change has been made as suggested.
- Line 65 “are included in a pattern dietary” correct. I think it is wrong presented
Change has been made as suggested.
- Line 104 “A study carried out in China in 2015“ I think this should be followed by past sentence “showed”
Change has been made as suggested.
- I can’t understand how could you estimate that the CV is related to the diet factors not due to any other causes including pregnancy “MedDiet and estimate CV risk based on diet”
The CV risk that we obtain from the results of the MedDietScore questionnaire only takes into account dietary factors in its calculation. That is why we specify that the CV risk presented is that attributable solely to diet. Overall CV risk will of course be modulated by other factors such as genetics or physical activity for example.
- How could you synchronize the risk of CV with the stage of the pregnancy in the tested groups especially who are at high risk?
The CV risk attributable to diet was calculated for the whole pregnancy as the information available on the women’s diet is their “average diet” during this period. When this information is collected, the women are asked to reflect their usual diet from conception up until that point.
- Also the presence of different races in another areas they got accustomed, may be other factor that should be taken in consideration
We understand that MedDiet calculates CV risk based on adherence to the Mediterranean pattern and that women of different origin may not be accustomed to eating following such pattern. We have now included this observation in the text. However, we do not believe this factor significantly affects our results given that the risk is calculated using each woman’s actual and current diet.
- The footnote of the table should precisely describe all abbreviations in the table
We have included all abbreviations in the footnote for clarity.
- Line 283: correct “val-ues”
Correction has been made.
- The P value should be written in the footnote.
We think this comment refers to Table 4. It appears that a column with the p-values was deleted during the formatting. We have included it once again and now all p-values are shown. If this is not what the reviewer was referring to, we will make the changes necessary.
- I think lines from 312- 315 need revision, I can’t get clear point.
We have rewritten these sentences for clarity.
- The association between cardiovascular risk and the presence of gestational diabetes, is not well addressed in the discussion and needs further recent citations.
We have expanded on this is the Discussion as requested including more recent citations.

Round 2
Reviewer 1 Report
Author's Notes
The authors undertook an interesting topic about the relationship between Mediterranean dietary pattern and cardiovascular risk in pregnant women.
We greatly appreciate the time and effort the reviewer has taken to help us improve our work. We hope to have addressed all their concerns accordingly but also remain open to receiving further feedback and making any additional clarifications or changes deemed necessary.
Note: In the marked version of the manuscript the references are not corrected as we have used the Track Changes function which makes using the Citation Manager in Word a little messy. We have, however, provided a clean version with the corrected references.
The Abstract section – it's written correctly, however, Simple Summary is redundant and not required in the Instruction for Authors.
We are not sure where the Simple Summary came from as we did not include it in the manuscript that we uploaded. We have found other numerous problems with the formatting changes made by the journal so perhaps there might be an issue there…
The Introduction section - The content of the Introduction section is not consistent. In verses 53-56, the Authors indicate that epidemiological studies on the preventive role of the Mediterranean diet in the context of cardiovascular diseases, give effects with long-term use and pregnancy lasts "only" nine months. On the other hand, the Authors indicate nutrients whose excess or deficiency may potentially affect the risk of pregnancy complications, although as mentioned earlier, the intake of individual nutrients does not reflect dietary patterns. Therefore, there is no link between the MedDiet nutritional pattern and the nutrients listed by the Authors.
We have revised this whole section and a great part of it has been rewritten taking the reviewer’s comments into account. We understand why the reviewer points out that effects of the MedDiet are seen after long-term use and pregnancy is a very limited period of time. However, it is a time in which most women will take special care of their nutrition, even if food aversions, nausea and other pregnancy symptoms sometimes affect food intake. The dietary pattern followed during this time should be reflective of their habitual long-term eating habits and may be even somewhat better, therefore, we feel that it is suitable for analysis in the context of the associations between MedDiet and its effects on CV health.
Please document with scientific evidence the continuation of the eating habits acquired during pregnancy by women later in life and enter this information in the Introduction section. The professional experience of the reviewer indicates that pregnant women return to their old eating habits from before pregnancy.
The Methods section - The recruitment process for the study would be clearer if it were presented in the form of a flowchart diagram.
We have included a clearer and more detailed explanation of the recruitment process but do not think that a flowchart diagram would add significant value. We would, however, include it if the reviewer deems it necessary.
In the opinion of the reviewer, the flowchart diagram would allow to show the full picture of the recruitment process and the problems that the researchers encountered at this stage.
In the Personal characteristics section, there is no information in which week of pregnancy the study participants were recruited.
Women were offered participation in this specific study during a post-birth visit but have been followed using the same care protocol since pregnancy confirmation before 12 weeks.
The above explanation by the authors contradicts the text in the manuscript „This study is an observational population-based period cross-sectional study using data from women who gave birth and had been attended throughout their pregnancy (from pregnancy confirmation before 12 weeks and with at least one visit per trimester).” - please decide whether participants were recruited before 12 weeks of pregnancy or at the postnatal visit.
The criteria for inclusion and exclusion from the study were also not specified.
The inclusion and exclusion criteria have been included in the previous section.
It is puzzling why no questions were asked about education, economic status and place of residence - these are the main factors determining food choices.
This information was collected but since it is not presented, it was not mentioned. We could add this information to Table 1 if the reviewer thinks it would add to our work.
In the opinion of the reviewer, these data are necessary for the presentation.
In verse 143, the authors provide "Maternal physical activity and employment status during the months of pregnancy were assessed" - it is not known what method was used to assess activity or to what period of time the question about physical activity referred to - the period before pregnancy or only during pregnancy - it is extremely difficult assess the status of physical activity, for example, in a pregnant woman with a high-risk pregnancy and a ban on any activity, who had high physical activity before pregnancy. There is no reference to the employment of pregnant women in any of the tables or in the subsequent text.
Physical activity during pregnancy was classified into the categories none, light and moderate using the self-reported information provided by the women. We understand that it is extremely difficult assess the status of physical activity especially given all the different situations the women may be in, however, as our aim is to assess CV risk due to diet it should not impact our results even if it would obviously have an impact on overall CV risk.
The authors still do not specify on what basis they assessed the physical activity of the study participants - was any questionnaire used for this purpose? What does it mean that a given pregnant woman had "none physical activity" or "light" or "moderate"?
It is true that we state that information on employment status during pregnancy was collected and then not presented. We could add this information, as well as that about education, economic status, and place of residence to Table 1 if the reviewer thinks it would add to our work.
If some data are not presented then the authors should not mention them in the text. Please consider whether this information (employment status) is significant in the context of other results or not - the answer will be an appropriate statistical analysis.
In the Dietary assessment section, the period covered by the questionnaire was not specified - only the period of pregnancy or the last year, was the questionnaire completed only once (in which trimester?) or once in each trimester? Was the questionnaire used validated in the group of pregnant women? - if not then it should not be used in this group. All the questions asked indicate the possibility of making several serious methodological errors.
Women were instructed to complete the validated MedDietScore questionnaire once, during their last programed antenatal visit (36-38 weeks), to reflect their habitual diet during the whole of their pregnancy.
The authors still have not answered the question whether the questionnaire used was validated on the pregnant population. Information on the time and frequency of the nutritional interview should be provided in the Dietary assessment section.
The Results sections - The authors repeated the same text in the Cardiovascular risk section (lines 188-191) and in the Food consumption of the Mediterranean diet pattern section (lines 206-208) and in the Outcomes section (lines 277-281) - in the last two cases as well as in verse 281 the sentences seem to be unfinished.
We have corrected this issue in the text, once again, we think this was an issue with the formatting.
It is puzzling why Table 2 does not include the results for alcoholic drinks when their consumption is shown in Table 1 and Table 4.
Table 2 is a reflection of habitual diet while Table 1 and Table 4 reflects the answers to the YES/NO question of if the women consumed alcohol at any point during pregnancy. All women answered Never (0 times/month) in the MedDietScore questionnaire, but some did say that at some point they had consumed alcohol. This could be for example, just one beer during all pregnancy. Given the answer to the MedDiet, question, that is why it was not presented but hawe have included it now.
If the answer is "never", this should be recorded in Table 2 as "100" and not "-".
The Discussion section - In verse 301, the authors refer to the MEDAS questionnaire - the skin should be explained.
MEDAS refers to the Mediterranean Diet Adherence Screener.
In the text of the Discussion section, the authors refer to studies on the analysis of pre-pregnancy eating behaviors, and in the presented study, the analysis was probably conducted only during pregnancy - these are incomparable periods.
We included information on the eating behaviors of the general adult Spanish population in order to provide context for the current situation regarding dietary behaviors in Spain but will removed it if the reviewer does not think it adds relevant information.
The reviewer is confident that the authors will find literature data describing patterns or eating habits of pregnant women from the Mediterranean region.
What is missing from this section is the comparison of dietary recommendations for Spanish pregnant women with the assumptions of the Mediterranean diet.
We have included a comparison among the two in the Discussion as suggested.
The Conclusions section - it is surprising why the Authors in the first sentence refer to the recommendations for pregnant women developed by the Spanish Society of Community Nutrition and not the MedDiet pattern.
We have rewritten the Discussion in order to better reflect the objective and results of this work.
All of the above comments indicate that the manuscript in its present form needs majore revision.

Author Response
We appreciate the time and effort the reviewer has dedicated to reviewing our work again. We hope to now have addressed all their concerns but still remain open to feedback and would be willing to make further changes if necessary.
The authors undertook an interesting topic about the relationship between Mediterranean dietary pattern and cardiovascular risk in pregnant women.
We greatly appreciate the time and effort the reviewer has taken to help us improve our work. We hope to have addressed all their concerns accordingly but also remain open to receiving further feedback and making any additional clarifications or changes deemed necessary.
Note: In the marked version of the manuscript the references are not corrected as we have used the Track Changes function which makes using the Citation Manager in Word a little messy. We have, however, provided a clean version with the corrected references.
The Abstract section – it's written correctly, however, Simple Summary is redundant and not required in the Instruction for Authors.
We are not sure where the Simple Summary came from as we did not include it in the manuscript that we uploaded. We have found other numerous problems with the formatting changes made by the journal so perhaps there might be an issue there…
The Introduction section - The content of the Introduction section is not consistent. In verses 53-56, the Authors indicate that epidemiological studies on the preventive role of the Mediterranean diet in the context of cardiovascular diseases, give effects with long-term use and pregnancy lasts "only" nine months. On the other hand, the Authors indicate nutrients whose excess or deficiency may potentially affect the risk of pregnancy complications, although as mentioned earlier, the intake of individual nutrients does not reflect dietary patterns. Therefore, there is no link between the MedDiet nutritional pattern and the nutrients listed by the Authors.
We have revised this whole section and a great part of it has been rewritten taking the reviewer’s comments into account. We understand why the reviewer points out that effects of the MedDiet are seen after long-term use and pregnancy is a very limited period of time. However, it is a time in which most women will take special care of their nutrition, even if food aversions, nausea and other pregnancy symptoms sometimes affect food intake. The dietary pattern followed during this time should be reflective of their habitual long-term eating habits and may be even somewhat better, therefore, we feel that it is suitable for analysis in the context of the associations between MedDiet and its effects on CV health.
Please document with scientific evidence the continuation of the eating habits acquired during pregnancy by women later in life and enter this information in the Introduction section. The professional experience of the reviewer indicates that pregnant women return to their old eating habits from before pregnancy.
I think we have not expressed ourselves correctly and there is some misunderstanding in what we are trying to convey. We agree with the reviewer that the usually healthier eating habits acquired in pregnancy are not normally continued after and that women return to their old eating habits from before pregnancy. However, as this work is centered on CV risk specifically during pregnancy, diet after pregnancy would not be relevant. At most, diet before pregnancy could be considered and that is why we mentioned that diet during pregnancy may be a reflection of long-term eating habits at their healthiest point and therefore when diet would be at its lowest effect as a CV risk factor.
The Methods section - The recruitment process for the study would be clearer if it were presented in the form of a flowchart diagram.
We have included a clearer and more detailed explanation of the recruitment process but do not think that a flowchart diagram would add significant value. We would, however, include it if the reviewer deems it necessary.
In the opinion of the reviewer, the flowchart diagram would allow to show the full picture of the recruitment process and the problems that the researchers encountered at this stage.
We have added the recruitment flowchart diagram as requested.
In the Personal characteristics section, there is no information in which week of pregnancy the study participants were recruited.
Women were offered participation in this specific study during a post-birth visit but have been followed using the same care protocol since pregnancy confirmation before 12 weeks.
The above explanation by the authors contradicts the text in the manuscript „This study is an observational population-based period cross-sectional study using data from women who gave birth and had been attended throughout their pregnancy (from pregnancy confirmation before 12 weeks and with at least one visit per trimester).” - please decide whether participants were recruited before 12 weeks of pregnancy or at the postnatal visit.
Eligible women were recruited and offered participation in the study at the postnatal visit but in order to be eligible they must have been followed in the same health department, to ensure the following of the same standard care protocol, from pregnancy confirmation before 12 weeks. Having been attended throughout pregnancy at the Valencia-La Fé Health Department and giving birth at the La Fé University and Polytechnic Hospital during 2020 and 2021 were two of the inclusion criteria established. We have rewritten this section to make the recruitment process more clear as well as included the flowchart as requested in the previous comment.
The criteria for inclusion and exclusion from the study were also not specified.
The inclusion and exclusion criteria have been included in the previous section.
It is puzzling why no questions were asked about education, economic status and place of residence - these are the main factors determining food choices.
This information was collected but since it is not presented, it was not mentioned. We could add this information to Table 1 if the reviewer thinks it would add to our work.
In the opinion of the reviewer, these data are necessary for the presentation.
We have included the information regarding education and place of residence in Table 1 as requested. No significant differences among the two groups were observed. Information on economic status has not been included as it is missing for a majority of the participants and therefore not reliable.
In verse 143, the authors provide "Maternal physical activity and employment status during the months of pregnancy were assessed" - it is not known what method was used to assess activity or to what period of time the question about physical activity referred to - the period before pregnancy or only during pregnancy - it is extremely difficult assess the status of physical activity, for example, in a pregnant woman with a high-risk pregnancy and a ban on any activity, who had high physical activity before pregnancy. There is no reference to the employment of pregnant women in any of the tables or in the subsequent text.
Physical activity during pregnancy was classified into the categories none, light and moderate using the self-reported information provided by the women. We understand that it is extremely difficult assess the status of physical activity especially given all the different situations the women may be in, however, as our aim is to assess CV risk due to diet it should not impact our results even if it would obviously have an impact on overall CV risk.
The authors still do not specify on what basis they assessed the physical activity of the study participants - was any questionnaire used for this purpose? What does it mean that a given pregnant woman had "none physical activity" or "light" or "moderate"?
Level of physical activity was self-reported and collected using a questionnaire that reproduced the corresponding questions from the Spanish National Health Survey which in turn are based on the International Physical Activity Questionnaire (IPAQ) which establishes the three levels of physical activity as:
Category 1: Low
This is the lowest level of physical activity. Those individuals who not meet criteria for categories 2 or 3 are considered low/inactive.
Category 2: Moderate
Any one of the following 3 criteria:
- 3 or more days of vigorous activity of at least 20 minutes per day OR
- 5 or more days of moderate-intensity activity or walking of at least 30 minutes per day OR
- 5 or more days of any combination of walking, moderate-intensity or vigorous intensity activities achieving a minimum of at least 600 MET-min/week.
Category 3: High
Any one of the following 2 criteria:
- Vigorous-intensity activity on at least 3 days and accumulating at least 1500 MET-minutes/
week OR
- 7 or more days of any combination of walking, moderate-intensity or vigorous intensity activities achieving a minimum of at least 3000 MET-minutes/week.
None of the women met the criteria for Category 3: High but we did have some totally sedentary individuals and that is why the categories presented in the text are None, Light and Moderate.
These details have been included in the text.
It is true that we state that information on employment status during pregnancy was collected and then not presented. We could add this information, as well as that about education, economic status, and place of residence to Table 1 if the reviewer thinks it would add to our work.
If some data are not presented then the authors should not mention them in the text. Please consider whether this information (employment status) is significant in the context of other results or not - the answer will be an appropriate statistical analysis.
We have included the information regarding employment status in Table 1 along with education, and place of residence. No significant differences among the two groups were observed.
In the Dietary assessment section, the period covered by the questionnaire was not specified - only the period of pregnancy or the last year, was the questionnaire completed only once (in which trimester?) or once in each trimester? Was the questionnaire used validated in the group of pregnant women? - if not then it should not be used in this group. All the questions asked indicate the possibility of making several serious methodological errors.
Women were instructed to complete the validated MedDietScore questionnaire once, during their last programed antenatal visit (36-38 weeks), to reflect their habitual diet during the whole of their pregnancy.
The authors still have not answered the question whether the questionnaire used was validated on the pregnant population.
The questionnaire by Panagiotakos or the questionnaire by Trichopoulou on which it is based have not been validated specifically for pregnant women, they have been validated for the general adult population and widely used in previous studies including studies on pregnant women. Some examples are:
Eckl, M. R., Brouwer-Brolsma, E. M., & Küpers, L. K. (2021). Maternal adherence to the mediterranean diet during pregnancy: a review of commonly used a priori Indexes. Nutrients, 13(2), 582.
Suárez-Martínez, C., Yagüe-Guirao, G., Santaella-Pascual, M., Peso-Echarri, P., Vioque, J., Morales, E., ... & NELA Study Group. (2021). Adherence to the mediterranean diet and determinants among pregnant women: the NELA cohort. Nutrients, 13(4), 1248.
Chatzi, L., Rifas‐Shiman, S. L., Georgiou, V., Joung, K. E., Koinaki, S., Chalkiadaki, G., ... & Oken, E. (2017). Adherence to the Mediterranean diet during pregnancy and offspring adiposity and cardiometabolic traits in childhood. Pediatric obesity, 12, 47-56.
Flor-Alemany, M., Migueles, J. H., Acosta-Manzano, P., Marín-Jiménez, N., Baena-García, L., & Aparicio, V. A. (2023). Assessing the Mediterranean diet adherence during pregnancy: Practical considerations based on the associations with cardiometabolic risk. Pregnancy Hypertension, 31, 17-24.
For questionnaires specifically validated in pregnant women as a Mediterranean diet index, we are only aware of the MDS-P that Mariscal-Arcas et al. (2009) proposed, which is a modification of the Trichopoulou questionnaire that also evaluates the adequacy of folic acid, Fe and Ca intake. We do not have this information on micronutrient intake and therefore could not apply it.
Mariscal-Arcas, M., Rivas, A., Monteagudo, C., Granada, A., Cerrillo, I., & Olea-Serrano, F. (2009). Proposal of a Mediterranean diet index for pregnant women. British journal of nutrition, 102(5), 744-749.
Information on the time and frequency of the nutritional interview should be provided in the Dietary assessment section.
We have included this information in the Dietary assessment section as requested.
The Results sections - The authors repeated the same text in the Cardiovascular risk section (lines 188-191) and in the Food consumption of the Mediterranean diet pattern section (lines 206-208) and in the Outcomes section (lines 277-281) - in the last two cases as well as in verse 281 the sentences seem to be unfinished.
We have corrected this issue in the text, once again, we think this was an issue with the formatting.
It is puzzling why Table 2 does not include the results for alcoholic drinks when their consumption is shown in Table 1 and Table 4.
Table 2 is a reflection of habitual diet while Table 1 and Table 4 reflects the answers to the YES/NO question of if the women consumed alcohol at any point during pregnancy. All women answered Never (0 times/month) in the MedDietScore questionnaire, but some did say that at some point they had consumed alcohol. This could be for example, just one beer during all pregnancy. Given the answer to the MedDiet, question, that is why it was not presented but have included it now.
If the answer is "never", this should be recorded in Table 2 as "100" and not "-".
Intake frequency of alcoholic drinks in Table 2 has been recorded as 100% in the Never category for both groups.
The Discussion section - In verse 301, the authors refer to the MEDAS questionnaire - the skin should be explained.
MEDAS refers to the Mediterranean Diet Adherence Screener.
In the text of the Discussion section, the authors refer to studies on the analysis of pre-pregnancy eating behaviors, and in the presented study, the analysis was probably conducted only during pregnancy - these are incomparable periods.
We included information on the eating behaviors of the general adult Spanish population in order to provide context for the current situation regarding dietary behaviors in Spain but will removed it if the reviewer does not think it adds relevant information.
The reviewer is confident that the authors will find literature data describing patterns or eating habits of pregnant women from the Mediterranean region.
In the Discussion we have retained the second paragraph which presents the adherence to MedDiet in the general Spanish population to provide the general context of the current situation in Spain and the third paragraph now presents solely information regarding MedDiet adherence rates for pregnant women in Spain. In the subsequent paragraphs we have included literature data pertaining specifically to the effects of MedDiet adherence in pregnant women.
What is missing from this section is the comparison of dietary recommendations for Spanish pregnant women with the assumptions of the Mediterranean diet.
We have included a comparison among the two in the Discussion as suggested.
The Conclusions section - it is surprising why the Authors in the first sentence refer to the recommendations for pregnant women developed by the Spanish Society of Community Nutrition and not the MedDiet pattern.
We have rewritten the Discussion in order to better reflect the objective and results of this work.
All of the above comments indicate that the manuscript in its present form needs major revision.
We appreciate the time and effort the reviewer has dedicated to reviewing our work again. We hope to now have addressed all their concerns but still remain open to feedback and would be willing to make further changes if necessary.
The authors undertook an interesting topic about the relationship between Mediterranean dietary pattern and cardiovascular risk in pregnant women.
We greatly appreciate the time and effort the reviewer has taken to help us improve our work. We hope to have addressed all their concerns accordingly but also remain open to receiving further feedback and making any additional clarifications or changes deemed necessary.
Note: In the marked version of the manuscript the references are not corrected as we have used the Track Changes function which makes using the Citation Manager in Word a little messy. We have, however, provided a clean version with the corrected references.
The Abstract section – it's written correctly, however, Simple Summary is redundant and not required in the Instruction for Authors.
We are not sure where the Simple Summary came from as we did not include it in the manuscript that we uploaded. We have found other numerous problems with the formatting changes made by the journal so perhaps there might be an issue there…
The Introduction section - The content of the Introduction section is not consistent. In verses 53-56, the Authors indicate that epidemiological studies on the preventive role of the Mediterranean diet in the context of cardiovascular diseases, give effects with long-term use and pregnancy lasts "only" nine months. On the other hand, the Authors indicate nutrients whose excess or deficiency may potentially affect the risk of pregnancy complications, although as mentioned earlier, the intake of individual nutrients does not reflect dietary patterns. Therefore, there is no link between the MedDiet nutritional pattern and the nutrients listed by the Authors.
We have revised this whole section and a great part of it has been rewritten taking the reviewer’s comments into account. We understand why the reviewer points out that effects of the MedDiet are seen after long-term use and pregnancy is a very limited period of time. However, it is a time in which most women will take special care of their nutrition, even if food aversions, nausea and other pregnancy symptoms sometimes affect food intake. The dietary pattern followed during this time should be reflective of their habitual long-term eating habits and may be even somewhat better, therefore, we feel that it is suitable for analysis in the context of the associations between MedDiet and its effects on CV health.
Please document with scientific evidence the continuation of the eating habits acquired during pregnancy by women later in life and enter this information in the Introduction section. The professional experience of the reviewer indicates that pregnant women return to their old eating habits from before pregnancy.
I think we have not expressed ourselves correctly and there is some misunderstanding in what we are trying to convey. We agree with the reviewer that the usually healthier eating habits acquired in pregnancy are not normally continued after and that women return to their old eating habits from before pregnancy. However, as this work is centered on CV risk specifically during pregnancy, diet after pregnancy would not be relevant. At most, diet before pregnancy could be considered and that is why we mentioned that diet during pregnancy may be a reflection of long-term eating habits at their healthiest point and therefore when diet would be at its lowest effect as a CV risk factor.
The Methods section - The recruitment process for the study would be clearer if it were presented in the form of a flowchart diagram.
We have included a clearer and more detailed explanation of the recruitment process but do not think that a flowchart diagram would add significant value. We would, however, include it if the reviewer deems it necessary.
In the opinion of the reviewer, the flowchart diagram would allow to show the full picture of the recruitment process and the problems that the researchers encountered at this stage.
We have added the recruitment flowchart diagram as requested.
In the Personal characteristics section, there is no information in which week of pregnancy the study participants were recruited.
Women were offered participation in this specific study during a post-birth visit but have been followed using the same care protocol since pregnancy confirmation before 12 weeks.
The above explanation by the authors contradicts the text in the manuscript „This study is an observational population-based period cross-sectional study using data from women who gave birth and had been attended throughout their pregnancy (from pregnancy confirmation before 12 weeks and with at least one visit per trimester).” - please decide whether participants were recruited before 12 weeks of pregnancy or at the postnatal visit.
Eligible women were recruited and offered participation in the study at the postnatal visit but in order to be eligible they must have been followed in the same health department, to ensure the following of the same standard care protocol, from pregnancy confirmation before 12 weeks. Having been attended throughout pregnancy at the Valencia-La Fé Health Department and giving birth at the La Fé University and Polytechnic Hospital during 2020 and 2021 were two of the inclusion criteria established. We have rewritten this section to make the recruitment process more clear as well as included the flowchart as requested in the previous comment.
The criteria for inclusion and exclusion from the study were also not specified.
The inclusion and exclusion criteria have been included in the previous section.
It is puzzling why no questions were asked about education, economic status and place of residence - these are the main factors determining food choices.
This information was collected but since it is not presented, it was not mentioned. We could add this information to Table 1 if the reviewer thinks it would add to our work.
In the opinion of the reviewer, these data are necessary for the presentation.
We have included the information regarding education and place of residence in Table 1 as requested. No significant differences among the two groups were observed. Information on economic status has not been included as it is missing for a majority of the participants and therefore not reliable.
In verse 143, the authors provide "Maternal physical activity and employment status during the months of pregnancy were assessed" - it is not known what method was used to assess activity or to what period of time the question about physical activity referred to - the period before pregnancy or only during pregnancy - it is extremely difficult assess the status of physical activity, for example, in a pregnant woman with a high-risk pregnancy and a ban on any activity, who had high physical activity before pregnancy. There is no reference to the employment of pregnant women in any of the tables or in the subsequent text.
Physical activity during pregnancy was classified into the categories none, light and moderate using the self-reported information provided by the women. We understand that it is extremely difficult assess the status of physical activity especially given all the different situations the women may be in, however, as our aim is to assess CV risk due to diet it should not impact our results even if it would obviously have an impact on overall CV risk.
The authors still do not specify on what basis they assessed the physical activity of the study participants - was any questionnaire used for this purpose? What does it mean that a given pregnant woman had "none physical activity" or "light" or "moderate"?
Level of physical activity was self-reported and collected using a questionnaire that reproduced the corresponding questions from the Spanish National Health Survey which in turn are based on the International Physical Activity Questionnaire (IPAQ) which establishes the three levels of physical activity as:
Category 1: Low
This is the lowest level of physical activity. Those individuals who not meet criteria for categories 2 or 3 are considered low/inactive.
Category 2: Moderate
Any one of the following 3 criteria:
- 3 or more days of vigorous activity of at least 20 minutes per day OR
- 5 or more days of moderate-intensity activity or walking of at least 30 minutes per day OR
- 5 or more days of any combination of walking, moderate-intensity or vigorous intensity activities achieving a minimum of at least 600 MET-min/week.
Category 3: High
Any one of the following 2 criteria:
- Vigorous-intensity activity on at least 3 days and accumulating at least 1500 MET-minutes/
week OR
- 7 or more days of any combination of walking, moderate-intensity or vigorous intensity activities achieving a minimum of at least 3000 MET-minutes/week.
None of the women met the criteria for Category 3: High but we did have some totally sedentary individuals and that is why the categories presented in the text are None, Light and Moderate.
These details have been included in the text.
It is true that we state that information on employment status during pregnancy was collected and then not presented. We could add this information, as well as that about education, economic status, and place of residence to Table 1 if the reviewer thinks it would add to our work.
If some data are not presented then the authors should not mention them in the text. Please consider whether this information (employment status) is significant in the context of other results or not - the answer will be an appropriate statistical analysis.
We have included the information regarding employment status in Table 1 along with education, and place of residence. No significant differences among the two groups were observed.
In the Dietary assessment section, the period covered by the questionnaire was not specified - only the period of pregnancy or the last year, was the questionnaire completed only once (in which trimester?) or once in each trimester? Was the questionnaire used validated in the group of pregnant women? - if not then it should not be used in this group. All the questions asked indicate the possibility of making several serious methodological errors.
Women were instructed to complete the validated MedDietScore questionnaire once, during their last programed antenatal visit (36-38 weeks), to reflect their habitual diet during the whole of their pregnancy.
The authors still have not answered the question whether the questionnaire used was validated on the pregnant population.
The questionnaire by Panagiotakos or the questionnaire by Trichopoulou on which it is based have not been validated specifically for pregnant women, they have been validated for the general adult population and widely used in previous studies including studies on pregnant women. Some examples are:
Eckl, M. R., Brouwer-Brolsma, E. M., & Küpers, L. K. (2021). Maternal adherence to the mediterranean diet during pregnancy: a review of commonly used a priori Indexes. Nutrients, 13(2), 582.
Suárez-Martínez, C., Yagüe-Guirao, G., Santaella-Pascual, M., Peso-Echarri, P., Vioque, J., Morales, E., ... & NELA Study Group. (2021). Adherence to the mediterranean diet and determinants among pregnant women: the NELA cohort. Nutrients, 13(4), 1248.
Chatzi, L., Rifas‐Shiman, S. L., Georgiou, V., Joung, K. E., Koinaki, S., Chalkiadaki, G., ... & Oken, E. (2017). Adherence to the Mediterranean diet during pregnancy and offspring adiposity and cardiometabolic traits in childhood. Pediatric obesity, 12, 47-56.
Flor-Alemany, M., Migueles, J. H., Acosta-Manzano, P., Marín-Jiménez, N., Baena-García, L., & Aparicio, V. A. (2023). Assessing the Mediterranean diet adherence during pregnancy: Practical considerations based on the associations with cardiometabolic risk. Pregnancy Hypertension, 31, 17-24.
For questionnaires specifically validated in pregnant women as a Mediterranean diet index, we are only aware of the MDS-P that Mariscal-Arcas et al. (2009) proposed, which is a modification of the Trichopoulou questionnaire that also evaluates the adequacy of folic acid, Fe and Ca intake. We do not have this information on micronutrient intake and therefore could not apply it.
Mariscal-Arcas, M., Rivas, A., Monteagudo, C., Granada, A., Cerrillo, I., & Olea-Serrano, F. (2009). Proposal of a Mediterranean diet index for pregnant women. British journal of nutrition, 102(5), 744-749.
Information on the time and frequency of the nutritional interview should be provided in the Dietary assessment section.
We have included this information in the Dietary assessment section as requested.
The Results sections - The authors repeated the same text in the Cardiovascular risk section (lines 188-191) and in the Food consumption of the Mediterranean diet pattern section (lines 206-208) and in the Outcomes section (lines 277-281) - in the last two cases as well as in verse 281 the sentences seem to be unfinished.
We have corrected this issue in the text, once again, we think this was an issue with the formatting.
It is puzzling why Table 2 does not include the results for alcoholic drinks when their consumption is shown in Table 1 and Table 4.
Table 2 is a reflection of habitual diet while Table 1 and Table 4 reflects the answers to the YES/NO question of if the women consumed alcohol at any point during pregnancy. All women answered Never (0 times/month) in the MedDietScore questionnaire, but some did say that at some point they had consumed alcohol. This could be for example, just one beer during all pregnancy. Given the answer to the MedDiet, question, that is why it was not presented but have included it now.
If the answer is "never", this should be recorded in Table 2 as "100" and not "-".
Intake frequency of alcoholic drinks in Table 2 has been recorded as 100% in the Never category for both groups.
The Discussion section - In verse 301, the authors refer to the MEDAS questionnaire - the skin should be explained.
MEDAS refers to the Mediterranean Diet Adherence Screener.
In the text of the Discussion section, the authors refer to studies on the analysis of pre-pregnancy eating behaviors, and in the presented study, the analysis was probably conducted only during pregnancy - these are incomparable periods.
We included information on the eating behaviors of the general adult Spanish population in order to provide context for the current situation regarding dietary behaviors in Spain but will removed it if the reviewer does not think it adds relevant information.
The reviewer is confident that the authors will find literature data describing patterns or eating habits of pregnant women from the Mediterranean region.
In the Discussion we have retained the second paragraph which presents the adherence to MedDiet in the general Spanish population to provide the general context of the current situation in Spain and the third paragraph now presents solely information regarding MedDiet adherence rates for pregnant women in Spain. In the subsequent paragraphs we have included literature data pertaining specifically to the effects of MedDiet adherence in pregnant women.
What is missing from this section is the comparison of dietary recommendations for Spanish pregnant women with the assumptions of the Mediterranean diet.
We have included a comparison among the two in the Discussion as suggested.
The Conclusions section - it is surprising why the Authors in the first sentence refer to the recommendations for pregnant women developed by the Spanish Society of Community Nutrition and not the MedDiet pattern.
We have rewritten the Discussion in order to better reflect the objective and results of this work.
All of the above comments indicate that the manuscript in its present form needs major revision.
Reviewer 3 Report
The authors worked adequately on the revised manuscript. Minor spelling mistakes are present and grammatical as well , please fix.
Also, I need to know how did the authors detected the significance for the presented data and why they are not applied in the tables of results?
Author Response
We appreciate the time and effort the reviewer has dedicated to reviewing our work again. We hope to now have addressed all their concerns but still remain open to feedback and would be willing to make further changes if necessary.
The authors worked adequately on the revised manuscript. Minor spelling mistakes are present and grammatical as well , please fix.
We have reviewed the manuscript for grammar and orthography as requested.
Also, I need to know how did the authors detected the significance for the presented data and why they are not applied in the tables of results?
The significance of the data was detected through the p-values which are presented in the tables.
Round 3
Reviewer 1 Report
The reviewer accepts the authors' explanations and corrections introduced by them in the text. I am only asking for the authors to supplement Limitation of the study with information that the MedDiet questionnaire was not validated on the pregnant population.